Initiating a watch list for Ebola virus antibody escape mutations

Miller Craig R. 1 2 3 4 crmiller@uidaho.edu
Johnson Erin L. 3
Burke Aran Z. 3 5
Martin Kyle P. 3 5
Miura Tanya A. 1 3
Wichman Holly A. 1 3 4
Brown Celeste J. 1 3 4
Ytreberg F. Marty 3 4 5 ytreberg@uidaho.edu
1 Department of Biological Sciences, University of Idaho , Moscow, ID , United States
2 Department of Mathematics, University of Idaho , Moscow, ID , United States
3 Center for Modeling Complex Interactions, University of Idaho , Moscow, ID , United States
4 Institute for Bioinformatics and Evolutionary Studies, University of Idaho , Moscow, ID , United States
5 Department of Physics, University of Idaho , Moscow, ID , United States
Diao Jiajie
Electronic publication date: 2016 Feb 16
Publication date: 2016
Volume: 4
Electronic Location ID: e1674
Received 2015 Dec 7; Accepted 2016 Jan 18
Copyright: ©2016 Miller et al.
Copyright year: 2016
Copyright holder: Miller et al.
License: This is an open access article distributed under the terms of the Creative Commons Attribution License, which permits unrestricted use, distribution, reproduction and adaptation in any medium and for any purpose provided that it is properly attributed. For attribution, the original author(s), title, publication source (PeerJ) and either DOI or URL of the article must be cited.
License URL: https://creativecommons.org/licenses/by/4.0/

Keywords: Ebola, Protein stability, FoldX, Molecular dynamics, Escape Mutations

Funding: National Science Foundation DEB1521049 National Institutes of Health P20 GM104420, P30 GM103324 Grant support for this research was provided by the National Science Foundation (DEB1521049) and the Center for Modeling Complex Interactions sponsored by the National Institutes of Health (P20 GM104420). Computer resources were provided in part by the Institute for Bioinformatics and Evolutionary Studies Computational Resources Core sponsored by the National Institutes of Health (P30 GM103324). The funders had no role in study design, data collection and analysis, decision to publish, or preparation of the manuscript.

==============================
The 2014 Ebola virus (EBOV) outbreak in West Africa is the largest in recorded history and resulted in over 11,000 deaths. It is essential that strategies for treatment and containment be developed to avoid future epidemics of this magnitude. With the development of vaccines and antibody-based therapies using the envelope glycoprotein (GP) of the 1976 Mayinga strain, one important strategy is to anticipate how the evolution of EBOV might compromise these efforts. In this study we have initiated a watch list of potential antibody escape mutations of EBOV by modeling interactions between GP and the antibody KZ52. The watch list was generated using molecular modeling to estimate stability changes due to mutation. Every possible mutation of GP was considered and the list was generated from those that are predicted to disrupt GP-KZ52 binding but not to disrupt the ability of GP to fold and to form trimers. The resulting watch list contains 34 mutations (one of which has already been seen in humans) at six sites in the GP2 subunit. Should mutations from the watch list appear and spread during an epidemic, it warrants attention as these mutations may reflect an evolutionary response from the virus that could reduce the effectiveness of interventions such as vaccination. However, this watch list is incomplete and emphasizes the need for more experimental structures of EBOV interacting with antibodies in order to expand the watch list to other epitopes. We hope that this work provokes experimental research on evolutionary escape in both Ebola and other viral pathogens.

Introduction

With nearly 30,000 confirmed cases and over 11,000 deaths, the recent Ebola virus (EBOV) epidemic in West Africa has dwarfed all recorded outbreaks of the disease (Centers for Disease Control, 2015). Now that the 2014 outbreak appears to be waning, it is critical to develop strategies for treatment and containment to avoid future epidemics of this magnitude. One important strategy is the development of vaccines. Two vaccines that express the EBOV envelope glycoprotein (GP) from the 1976 Mayinga strain are in phase III clinical trials: rVSV-ZEBOV and ChAd3-ZEBOV (Garbutt et al., 2004; Stanley et al., 2014; Marzi et al., 2013). Early reports suggest that rVSV-ZEBOV is highly effective at preventing EBOV infection (Henao-Restrepo et al., 2015). A related strategy is antibody-based therapeutics. For example, ZMapp has been shown to be effective in treating non-human primates and has been used to treat small numbers of humans with Ebola (Qiu et al., 2014; Borio, Cox & Lurie, 2015). The monoclonal antibodies in ZMapp were generated by vaccination of mice with GP from the 1976 Mayinga strain (Wilson et al., 2000; Qiu et al., 2011; Qiu et al., 2014).

A key course of action to prepare for future EBOV outbreaks is to anticipate how the evolution of antibody escape mutants in the virus might compromise treatment efforts. Antibody escape mutants have arisen in the laboratory when recombinant vesicular stomatitis viruses expressing the GP protein of EBOV or Marburg virus were grown in the presence of anti-GP antibodies (Kajihara et al., 2013). In that study, a single amino acid substitution conferred viral resistance to the antibodies. Similarily, a single amino acid change in GP of the EBOV Kikwit 95 strain in a macaque treated with monoclonal antibodies resulted in fatal infection (Qiu et al., 2012). Mutational changes in GP have also been found to impact immune responses to the virus; substitutions at N-linked glycosylation sites can alter antigenicity and immunogenicity, in some cases preventing binding to the KZ52 antibody (Dowling et al., 2007; Lennemann et al., 2015). Antibody escape mutants are also known in influenza A, HIV 1, measles and respiratory syncytial virus infections (Smith et al., 2004; Geiß& Dietrich, 2014; Schrag, Rota & Bellini, 1999; Zhao et al., 2006).

Sequencing studies have shown that there is a high level of genetic variation in EBOV and that GP has the largest variation among EBOV proteins (Gire et al., 2014; Tong et al., 2015; Park et al., 2015). As of August 2015, sequences from the 2014 outbreak show that 106 of the 676 sites in GP experienced a mutation and the strains differ from the 1976 Mayinga strain used in developing interventions by an average of 20.2 nucleotide changes. Thus, there is a very real possibility of antibody escape mutants arising in EBOV GP. A recent study found that none of the genetic changes have altered the function of the virus (Olabode et al., 2015). However, they did not consider interactions with antibodies or implications of unobserved mutations.

The purpose of this study is to initiate a watch list of potential antibody escape mutants for the EBOV GP. We focus on the KZ52 antibody as it is one of the few with an available structure bound to EBOV GP. KZ52 has virus neutralization activity in vitro and protects guinea pigs from EBOV disease (Maruyama et al., 1999; Parren et al., 2002). Although KZ52 does not protect non-human primates from EBOV disease (Oswald et al., 2007), it was originally isolated from the blood of human EBOV survivors (Maruyama et al., 1999). Using the experimental structure of the Zaire EBOV GP bound to antibody KZ52 (Fig. 1) (Lee et al., 2008), we performed molecular modeling to estimate the folding and binding stabilities for every possible amino acid mutation of GP. Our approach is general and could be applied to other EBOV epitopes, or other viruses, as experimental structures become available. We emphasize from the outset that this is an in silico study aimed at identifying mutations with an increased risk of escaping immune response; our intention is to provoke experimental research on evolutionary escape in both Ebola and other viral pathogens.

Figure 1 Structure of Ebola glycoprotein trimer in complex with the KZ52 antibody as viewed from the side (A) and the bottom (B).

GP1 is in gray, GP2 is in yellow and the structure is after 10 ns of MD simulation. The six watch list sites that are predicted to contain antibody escape mutants are shown as red spheres and are all located in GP2 (Table 2 and Fig. 2.)

Methods

Overview

To initiate a watch list for the Ebola virus (EBOV) glycoprotein (GP) it is necessary to determine how amino acid mutations modify stabilities for GP folding, forming a trimer and binding to the KZ52 antibody. That is, we need to calculate ΔΔG values for binding and folding. Ideally, these calculations would be performed using a statistical-mechanics-based method such as we have done previously (Lee et al., 2011; Zhan & Ytreberg, 2015). However, such methods are computationally expensive and are not feasible for the current study where it was necessary to calculate 25,840 values of ΔΔG (340 residues × 19 possible mutations to other residues × 4 types of stability calculations). Instead, we decided to use a semi-empirical method for calculating ΔΔG values. Because online-only software was not practical given the large number of mutations, we chose to use the software FoldX (Schymkowitz et al., 2005; Guerois, Nielsen & Serrano, 2002). FoldX can be run in parallel on a computer cluster since the binary is available.

We hypothesized that because protein structures are not static, improvements in ΔΔG estimation might be achieved by using molecular dynamics simulation to sample the configurational space for the proteins and then analyze snapshots from these simulations in FoldX. We selected 20 test systems (10 folding and 10 binding) to assess whether this strategy improves estimation of experimental stability data. In the Supplemental Information, we describe our criteria for selecting test systems and then show that using 100 molecular dynamics snapshots and averaging the FoldX results provides more accurate estimates of ΔΔG as compared to using FoldX on a single experimental structure. The molecular dynamics plus FoldX methodology we used on the test systems was identically applied to the Ebola system. After explaining how structures were prepared and arranged, we describe this methodology in the subsections below.

Stability estimation

Structure preparation

Preparation of the test system structures is described in the Supplemental Information. For EBOV GP, the amino acid sequence was based on the 1976 Mayinga strain obtained from GenBank accession number AF086833. We downloaded PDB accession number 3CSY as our template structure. The file 3csy.pdb was modified to remove all but one copy each of GP1, GP2, antibody light chain and antibody heavy chain (one third of the GP-KZ52 trimeric complex). SWISS-MODEL was then used to generate structures for each of the four chains using 3csy.pdb as a template Arnold et al. (2005). The experimental structure 3csy has missing residues 190-213 that are predicted to be intrinsically disordered but SWISS-MODEL incorrectly generated helical structures for these residues. Thus, we removed residues 190-213 from the SWISS-MODEL structure and used MODELLER to rebuild the coordinates of the missing residues Sali & Blundell (1993). The resulting structure had no secondary structure content in residues 190-213. The full trimeric complex was then created using the symexp command in PyMOL. The final trimer structure (see Fig. 1) contains three copies each of residues 32-276 for GP1, residues 503-597 for GP2, residues 1-225 for KZ52 heavy chain and residues 1-216 for KZ52 light chain.

System configuration

Arrangement of the test systems is described in the Supplemental Information. EBOV GP was configured as four systems: (i) unbound GP1, (ii) unbound GP2, (iii) trimer consisting of three copies of GP1 and GP2 and (iv) antibody complex consisting of three copies each of GP1, GP2 and the KZ52 antibody. Snapshots from systems (i) and (ii) were used to estimate mutational effects on folding stability of the unbound proteins GP1 and GP2, respectively. Snapshots from (iii) were used to estimate the affinity of GP1–GP2 (dimer bind). This was done by calculating the affinity for all three copies of GP1 binding to GP2 and then dividing this value by three. Snapshots from (iii) were also used to estimate the affinity for GP1–GP2 dimers binding to one another (trimer bind). This was done by calculating the affinity for one GP1–GP2 dimer binding to the other two dimers. Finally, snapshots from (iv) were used to estimate the GP-KZ52 affinity by calculating the affinity of all of the GP1–GP2 dimers to their corresponding KZ52 antibodies and dividing this value by three.

Molecular dynamics simulations

The software package GROMACS 5.0.3 was used for all MD simulations with the Charmm22* forcefield (Hess et al., 2008). The system was placed in a dodecahedral box of TIP3P water and given neutral charge by adding Na+ and Cl− ions at a concentration of 0.15 mol/L. Each system was then minimized using steepest decent for 1,000 steps. To allow for some equilibration of the water around the proteins, each system was then simulated for 1 ns with the positions of all heavy atoms in the complex harmonically restrained, and then simulated for another 1 ns with no restraints. During the restrained simulations the temperature of the system was increased linearly from 100 K to 300 K for the test systems and to 310 K for the EBOV GP systems and the pressure was maintained at 1 atm using the Berendsen algorithm. Production simulations for each system were then carried out for 100 ns with pressure maintained using Parrinello-Rahman coupling. For all simulations, the LINCS algorithm was used to constrain all bonds to their ideal lengths and virtual sites were used allowing the use of a 5 fs timestep. The temperature was controlled using the v-rescale option. Particle mesh Ewald was used for electrostatics with a real-space cutoff of 1.2 nm. Van der Waals interactions were cut off at 1.2 nm with the Potential-shift-Verlet method for smoothing interactions. During the 100 ns production simulation snapshots were saved every 100 ps giving 100 snapshots for each system.

FoldX

Each of the 100 snapshots captured during MD simulations was then analyzed using FoldX (Schymkowitz et al., 2005; Guerois, Nielsen & Serrano, 2002). We initially minimized structures six times in succession using the RepairPDB command to obtain convergence of the potential energy. All single amino acid mutations were then generated using BuildModel. Finally, protein folding stabilities were estimated using Stability on the monomer structures and binding stabilities were estimated using AnalyseComplex on the protein complexes. For each mutation we then estimated ΔΔG by averaging across all 100 individual snapshots estimates.

Thresholds for functionality and antibody disruption

To define the range of stability change where the GP protein is likely to remain functional, we began by noting that in previous work on the bacteriophage ϕX174 (Miller et al., 2014), 77 of 79 (97.5%) of observed functional mutations have estimated stability effects on both folding and binding in the range −2.5 < ΔΔG < 2.5 kcal/mol. The large amount of available Ebola sequences allows us to survey a set of presumably functional mutations in Ebola and ask how many of these are categorized as functional vs non-functional using this preliminary criteria. We downloaded 922 sequences from the NCBI Virus Variation Ebolavirus Database on August 20, 2015 (Brister et al., 2013; National Center for Biotechnology Information, 2015) (Species = Zaire ebolavirus, Host = Any, Region = Any, Genome Region = Spike glycoprotein). To this set we appended 39 sequences from Leroy et al. (2004) and Wittmann et al. (2007) along with the two escape mutations described in Qiu et al. (2012). We compared all 963 sequences to our reference sequence, GenBank Accession AF086833, and thereby identified 41 mutational differences (Table 1) within the structured regions modeled here. Four of the 41 mutations (9.8%) have a functional stability effect (i.e., ΔΔG for monomer folding, dimer binding or trimer binding) that falls outside the ±2.5 zone. Because our objective is to limit the rate of false exclusions to ≤5%, we expanded the functional zone to ±3.0. This shifts two of the mutations back into the functional zone, leaving 2 of 41 (4.9%) predicted to be non-functional.

Table 1 Model predicted effects on stability of 41 observed mutations in EBOV GP.

The one observed mutation that is also on the watch list is indicated in red. The two mutations that our methods falsely excludes as non-functional are indicated in blue. All numerical entries are ΔΔG values in units of kcal/mol.

Mutation	Antibody binda	Dimer bindb	Trimer bindc	Monomer foldd	
N107D	0.00	−0.09	0.00	1.31	
L111F	0.00	0.00	0.00	2.31	
I129V	0.00	0.05	0.02	0.77	
D150A	0.00	0.00	0.01	0.6	
D163N	0.00	1.12	0.57	0.39	
I170L	0.00	0.01	0.02	2.31	
I170F	0.00	0.03	0.05	16.48	
V181I	0.00	0.05	0.00	−0.73	
T206M	−0.30	−0.33	0.01	−0.14	
G212D	0.00	0.19	−0.04	−0.11	
Y213H	0.00	0.41	−0.01	1.27	
Y214H	0.00	−0.01	0.00	0.18	
T216P	0.00	−0.01	0.00	2.27	
R219K	0.00	0.00	0.00	0.00	
A222V	0.00	0.00	0.00	−0.11	
E229K	0.00	0.00	0.00	−0.17	
T230A	0.00	0.00	0.00	0.62	
T240N	0.00	0.00	0.00	0.86	
S246P	0.00	0.00	0.00	−1.11	
L254I	0.00	0.00	0.00	0.81	
L254V	0.00	0.00	0.00	1.39	
Q255R	0.00	0.00	0.00	0.1	
I260R	0	0	0	1.78	
T262A	0	0	0	−0.08	
W275L	0	0	0	0.09	
A503V	−0.17	0.09	0	0.1	
Q508R	0.16	−0.03	0	0.54	
Y517C	0.01	0.26	0.01	1.38	
G524D	−0.01	0.14	2.31	2.12	
A526T	0.00	0.02	0.56	0.80	
I527T	0.00	0.18	0.15	1.04	
P537L	0.00	0.23	0.42	0.53	
I544T	0.00	0.36	−0.01	0.47	
E545D	0.00	0.5	0.00	0.46	
N550K	4.59	0.01	0.00	0.62	
D552N	1.76	0.23	0.00	0.13	
A562D	−0.06	2.98	0.02	0.67	
L571R	0.00	0.05	2.34	0.27	
L573R	0.00	2.78	−0.25	1.30	
W597F	0.00	0.07	0.48	−0.11	
W597C	0.00	0.35	3.51	0.26	
Notes.

a Binding affinity between GP and the KZ52 antibody.

b Binding affinity between GP1 and GP2.

c Binding affinity between three GP1-GP2 dimers.

d Folding stability for GP2.

It is worth noting that the observed incidence of two false exclusions in a sample of 41 is consistent with our method having predictive power to distinguish functional from non-functional mutations. Of the 6,460 possible mutations for GP, our method categorizes 5,303 (82.1%) as functional and 1,157 (17.9%) as non-functional. If our method lacked predictive power we would expect a random sample of 41 mutations to contain 33.7 functional and 7.3 non-functional mutants. The binomial probability that such a random sample would contain ≤2 non-functional proteins by chance is 0.018. Unfortunately, because we lack a list of known non-functional mutations, we cannot perform the converse test and ask what proportion of non-functional mutations does our method correctly identify as such.

Table 2 Watch list mutations and their effects on stability.

All numerical entries are ΔΔG values in units of kcal/mol. Binding affinity results for forming the GP trimer are all zero and are not shown. The one observed mutation on the watch list is indicated in red.

GP2 mutationa	Antibody bindb	Dimer bindc	Monomer foldd	
N506W	3.40	0.04	−0.41	
N506Y	2.56	0.09	−0.55	
P513H	2.52	0.01	0.95	
P513W	2.19	0.01	0.86	
N550Q	3.76	0.01	0.92	
N550K	4.59	0.01	0.62	
N550P	3.82	0.14	2.20	
N550F	10.01	0.03	2.09	
N550H	5.50	0.03	1.81	
N550I	5.28	0.02	1.66	
N550E	3.49	−0.04	1.14	
N550R	5.34	−0.03	0.98	
N550W	13.52	0.02	2.29	
N550V	2.08	0.02	1.74	
N550Y	13.52	0.04	1.98	
N550M	3.29	0.02	−0.15	
D552S	2.10	0.75	0.33	
D552Q	2.19	0.36	0.29	
D552K	2.61	0.28	0.16	
D552T	2.40	1.01	1.47	
D552F	4.11	0.43	0.14	
D552A	2.17	0.71	0.48	
D552H	4.53	0.55	0.29	
D552G	2.61	0.39	0.01	
D552R	3.30	0.26	0.34	
D552W	5.05	0.41	0.61	
D552V	2.41	0.75	1.95	
D552Y	4.71	0.42	0.13	
G553M	8.77	−0.01	2.94	
G557F	2.26	0.13	−1.34	
G557H	3.72	0.67	−0.05	
G557R	2.29	0.17	−0.62	
G557W	3.21	0.69	−1.32	
G557Y	2.81	0.14	−1.19	
Notes.

a The 34 mutations are distributed among six sites in GP2.

b Binding affinity between GP and the KZ52 antibody.

c Binding affinity between GP1 and GP2.

d Folding stability for GP2.

Figure 2 Watch list mutations disrupt KZ52 antibody binding but not GP folding and trimer formation.

For each possible GP mutation, only the maximum of folding stability, dimer binding stability (interaction of GP1 and GP2) or trimer binding stability (interaction of a GP1-GP2 dimer with other dimers) is plotted on the y-axis. Symbols in the inset legend indicate which of the three is plotted. The GP-KZ52 binding affinity is plotted on the x-axis. Mutations with x-axis values −3 < ΔΔG < 3 kcal/mol are considered functional since they are likely to retain the ability to fold and form trimers (regions A and D). Mutations with y-axis values ΔΔG > 2 kcal/mol have the potential to disrupt antibody binding (regions C and D). The watch list mutations (region D) are those that are likely to be both functional and disrupt antibody binding. The reasoning behind using a different cutoff for functional as compared to antibody binding is described in the main text.

How sensitive is the size of the watch list to the rate of false exclusions? The following argument suggests that even if the false exclusion rate could be reduced to zero, it would have a very small effect on the watch list. The application of a functional zone bewteen ±3.0 kcal/mol along with an antibody disruption criteria of ΔΔG > 2.0 kcal/mol leads to a watch list of 34 mutations. Of the 6,460 possible mutations, our method categorizes 1,157 as non-functional. If 5% of these are actually functional, it suggests that we have omitted approximately 6,460(0.05) = 58 mutations from the set of functional mutations. However, very few of these would likely disrupt antibody binding. Among all 6,460 mutations, 66 (or ≈1%) are identified as disrupting antibody binding. Assuming false exclusion is independent of antibody disruption, we would expect that 58(0.01) = 0.6, or less than one mutation being falsely omitted from the watch list.

Results and Discussion

We identified potential antibody escape mutations for the watch list by considering every possible GP mutation and finding those that disrupt binding between GP and KZ52 but do not disrupt the ability of GP to fold and form a complex. The GP protein is cleaved into two subunits, GP1 and GP2, and the final structure is a trimer consisting of three GP1–GP2 dimers (Fig. 1). We used a combination of molecular dynamics and FoldX (Schymkowitz et al., 2005; Guerois, Nielsen & Serrano, 2002) because preliminary analysis of 20 test systems showed that combining these methods improved our ability to predict experimental results (see Supplemental Information). To our knowledge, this method has not been used in previous studies.

Our conceptual approach to creating a watch list is to identify mutations that are both functional and disrupt antibody binding. We therefore sought to remove mutations that are non-functional and, from those that remain, identify the ones that disrupt antibody binding. The function of GP is to mediate viral entry into the cell. There are multiple ways mutation can disrupt this function. For example, studies have shown that mutations in GP can reduce infectivity (Ito et al., 1999; Watanabe et al., 2000; Davidson et al., 2015), transduction and host cell binding (Dube et al., 2009; Brindley et al., 2007). Another way to be non-functional is for a mutation to render GP unable to fold and bind together to form a stable complex. Here we focus on this stability aspect of functionality and remove those mutations our model predicts will not fold or form a complex. It is important to appreciate that our approach is conservatively inclusive: if we could remove all non-functional mutations instead of the subset identified as unstable, the watch list would be reduced in size.

Identifying mutations that disrupt antibody binding but not the ability to fold and bind into a functional complex requires defining thresholds on changes in stability (ΔΔG) for both criteria. These criteria should be conservative to reduce exclusion of mutations that could compromise treatment efficacy from the watch list. For functionality, previous work on a coat protein in a different virus (Miller et al., 2014) indicated that the stability effect of virtually all observed mutations is in the range of −2.5 < ΔΔG < 2.5 kcal/mol. To determine if those criteria also hold for EBOV GP, we compared 963 available sequences of GP, identified 41 mutations in the structured regions that have arisen in natural or lab populations, and found that four of the 41 (9.8%) were classified as non-functional. To be conservative, we expanded the functional zone to −3.0 < ΔΔG < 3.0 kcal/mol. This functional threshold is more inclusive and reduces our error rate to below 5%: two of the 41 mutations (4.9%) are falsely classified as non-functional (Table 1). As we reason in the Methods, even if the false exclusion rate could be driven to zero, we expect it would change our watch list very little. For disruption of antibody binding, we used a threshold of ΔΔG > 2.0 kcal/mol. This was based on refining our preliminary threshold by 0.5 kcal/mol, but in the opposite direction so as to be more inclusive. The implications of this threshold choice and alternatives to it will be discussed below.

Figure 2 provides a graphical illustration of how mutations were selected to be on the watch list. The maximum functional stability for all mutations is plotted against the corresponding change in the antibody binding affinity. The 34 mutations in the lower right quadrant are those that belong on the watch list since they are classified as both functional and disruptive to antibody binding. The specific mutations on the watch list are given in Table 2. If any of the mutations in this table appear in a real population, it indicates an increased risk of escaping the normal immune response. One of these mutations (N550K) has already appeared in humans thought to have been infected by gorillas in Central Africa between 2001 and 2003 (Leroy et al., 2004). This mutation is present in all sequenced isolates from that outbreak.

In contrast to constructing a simple list of all possible mutations near an epitope, the watch list in Table 2 is quite specific. The 34 watch list mutations are concentrated at just six residues and all of these lie at the interface between GP2 and KZ52, as one might intuitively expect from the structure (Fig. 1). Yet, most mutations of GP sites that are within four angstroms of the KZ52 antibody are not predicted to disrupt antibody binding. Only six of the 23 (26%) interface sites and 34 of 437 (7.8%) of the possible mutations at these sites are on the watch list.

In order to facilitate use of other possible criteria and/or thresholds, the Supplemental Information includes a sortable and searchable spreadsheet with all 6,460 mutations of GP. We provide this spreadsheet because we recognize that the relationship between antibody binding affinity and the ability of the antibody to neutralize EBOV is not well understood. Work in influenza suggests that as affinity decreases, the ability of an antibody to neutralize a virus decreases rapidly and in a non-linear fashion (Kostolanskỳ et al., 2000). This makes intuitive sense because the relationship between the change in affinity and the change in the ratio of bound to unbound antibody is nonlinear; ΔΔG values of 1.0, 1.5 and 2.0 kcal/mol correspond to changing the ratio from 19 to 8.1 to 3.5% of its original value. The size of the watch list depends on how we define the threshold for antibody binding (Fig. 2). If the threshold is lowered from 2.0 to 1.5 to 1.0, the watch list grows from 34 to 49 to 73 mutations. This highlights the need for more experimental studies that assess how disrupting antibody binding influences immune response.

Davidson et al. (2015) recently conducted an alanine-scanning mutagenesis study on GP that can be qualitatively compared to our work. Specifically, they individually mutated each residue of the GP protein to alanine and measured changes in GP-KZ52 binding affinities relative to the unmutated form. They identified five residues that are critical for KZ52 antibody binding: C511, N550, D552, G553, and C556. Three of these sites are found on our watch list in Table 2 (N550, D552, and G553) and 25 of the 34 (74%) watch list mutations are found at these three sites. For the two other critical residues identified by Davidson et al. (2015) (C511 and C556), our results agree that antibody binding is disrupted by mutations at these sites, but we estimate that folding is also disrupted, and hence the exclusion from the watch list. If we ignore our criteria that mutations do not disrupt folding stability or the formation of dimers and trimers, we identify eight residues where at least one mutation will disrupt KZ52 antibody binding: N506, C511, P513, N550, D552, G553, C556, G557 (all individual mutations can be obtained from the spreadsheet in the Supplemental Information). Overall, we conclude that our results are generally consistent with the findings of Davidson et al. (2015).

The watch list remains incomplete and putative for several reasons. First, although our list was generated for one EBOV epitope and its interactions with the KZ52 antibody, it is known that there are multiple epitopes (Fig. 3). Indeed, a recent study found mutations of a conserved threonine in the EBOV mucin-like domain that is required for protection by the 14G7 antibody (Park et al., 2015). This highlights the need for more experimental structures of antibodies interacting with viral proteins. With more experimental structures, it would be possible to expand the watch list to incorporate more epitopes. Second, the watch list only includes substitutions that are predicted to individually disrupt antibody binding while remaining functional. It is alternatively possible that immune escape could arise by the accumulation of several changes, each of modest stability effect but with a large cumulative effect on antibody binding. How multiple substitutions interact to produce cumulative effects on stability is not well understood and is an important consideration for future studies. Third, the watch list has not been experimental validated (except in its general consistency with the work of Davidson et al. (2015)) either in terms of mutational effects on GP folding and binding affinities, nor on the downstream immune system consequences. Our hope is that this work will motivate such research.

Figure 3 Structure of Ebola GP1-GP2 trimer complex (A) and individual GP1-GP2 dimer (B) with structural epitopes from KZ52 and other known linear epitopes.

KZ52 is in green, other known linear epitopes are in blue (Becquart et al., 2014). The watch list generated for the current study is for the green region only, since structures are required for the method used, highlighting the need for more experimental structures of Ebola with antibodies.

In summary, we have initiated a watch list of potential antibody escape mutations of EBOV by considering the interactions between GP and antibody KZ52. This initial watch list contains 34 mutations in six sites in GP2, and one of these mutations (N550K) was seen in humans in a previous outbreak. We believe initiating a watch list is an important first step to predicting how the evolution of EBOV could undermine treatment efforts. Our intention is that the watch list motives experimental research testing the strategy we have employed. This study further emphasizes the need for more experimental structures of antibodies interacting with EBOV in order to produce a comprehensive watch list. We highlight the need for ongoing monitoring of EBOV sequences in human outbreaks. If mutations on the watch list appear in human populations infected by EBOV, treatment with vaccines or antibody therapies may be compromised. Furthermore, if mutations from the watch list arise and increase in frequency within an immunized population, it would suggest that the virus is responding to selective pressure exerted by the vaccine. Monitoring will be much more powerful as the watch list is expanded and experimentally validated. Finally, we suggest that the approach used here is general and could be applied to other viruses for which experimental structures are available.

Supplemental Information

Supplemental Information 1 Supplemental Material

Click here for additional data file.

Supplemental Information 2 Workflow_Notebook

Click here for additional data file.

Supplemental Information 3 Compressed folder with workflow notebook, scripts, code, and raw input files

Click here for additional data file.

Supplemental Information 4 Supplemental Data: excel spreadsheet with estimated stability effects of all 6,460 mutations

Click here for additional data file.

Additional Information and Declarations

Competing Interests

Author Contributions

Data Availability

The authors declare there are no competing interests.

Craig R. Miller and F. Marty Ytreberg conceived and designed the experiments, performed the experiments, analyzed the data, contributed reagents/materials/analysis tools, wrote the paper, prepared figures and/or tables, reviewed drafts of the paper.

Erin L. Johnson, Aran Z. Burke and Kyle P. Martin wrote the paper, reviewed drafts of the paper.

Tanya A. Miura, Holly A. Wichman and Celeste J. Brown conceived and designed the experiments, wrote the paper, reviewed drafts of the paper.

The following information was supplied regarding data availability:

All associated code and files have been bundled into a Workflow_Files Folder that is available as Supplemental Information.

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
