# Peer review of "Initiating a watch list for Ebola virus antibody escape mutations"

_PeerJ, doi:10.7717/peerj.1674_

## Round 0.1 · original submission · Major Revisions

· Academic Editor

Major Revisions

As you can read from reviewers' comments, the experimental proof is essential for this study.

Please consider, and address, the comments of all the reviewers.

·

Basic reporting

No comments

Experimental design

More discussion is needed for the comparison the method author used and other method, such as the difference between FoldX and Rossetta for the snapshot capture.

Validity of the findings

The validation is not enough. It will be great if author can confirm the experiment result using in vitro binding assay.

Comments for the author

The manuscript by Ytreberg and coworkers describes a Ebola virus escape mutation analysis. Molecular modeling was used based on the published crystal structure. The result is reliable. Overall, this is a technically sound study and is written with clarity.
However, there are a few concerns that should be addressed prior to publication.

By the way,
In line 43: what is the meaning of “average of 20.2 changes”

Reviewer 2 ·

Basic reporting

No comments

Experimental design

No comments

Validity of the findings

No comments

Comments for the author

Antibody escape mutation is an evolutionary strategy of viral pathogens to survive neutralizing antibody treatment. Prediction of potential escape mutations before they prevail in patients is important to disease control. In this study, the authors generated a watch list of potential escape mutations of the EBOV virus against a neutralizing antibody KZ52 by modeling the interactions between GP and the antibody using a combination of Molecular Dynamics and FoldX. Mutations that are predicted to disrupt GP-Antibody interaction but not the ability of GP to fold or dimerize were included in the list. This approach generated a list containing 34 mutations at six sites in GP2.

This is a timely and potentially important study in the background of the recent outbreak of Ebora epidemic in west Africa. However, due to a complete lack of experimental proof, the validity of the watch list is still questionable. Ideally, the authors should test at least a couple mutations for their impacts on antibody binding to confirm their predictions. The author should also discuss their findings in the context of a recently published study in which a comprehensive alanine scan were performed on GP of EBOV for antibody escape (Davidson et al. 2015 J Virol. 1:89:10982-92).

The criteria the authors used to score potential escaping site, “i.e. mutation would disrupt GP2-antibody interaction but does not alter the ability of GP to fold or dimerize” is also questionable. Even when a mutation retained the WT folding and dimerization ability, it is hard to conclude that this mutation would not interfere with viral infectivity in another way. Is there any evidence that mutations in the antibody binding interface of GP2 have little functional significance? As a matter of fact, the Davidson et al. work has found that mutations of residue 550, 552 and 553 reduced viral infectivity (ref). How many of the 106 natural occurring mutation sites reside in the antibody (KZ52) binding interface? Perhaps those sites are the beginning pool of antibody escaping mutation since they are functionally neutral.

In the crystal structure, the G557 residue is distal to the antibody and does not form any contact with antibody residues. Can the authors speculate how G557 mutants would possibly disrupt antibody binding while maintain proper folding?

Reviewer 3 ·

Basic reporting

The manuscript entitled “Initiating a watch list for Ebola virus antibody escape mutations" reports a watch list of potential antibody escape mutants for the Ebola glycoprotein using a combination of molecular dynamics and FoldX as molecular modeling. This approach is general and helpful for other viruses. The article is well written and the structure of the article is consistent with the journal requirements. The data has been made available in accordance with PeerJ guidelines. However the figures are not well presented.

Experimental design

This submission is within the scope of the journal, and the method in this paper is well presented and has potential merit in expanding to other viruses. However, some details in how the theoretical basis of calculating ΔΔG values is missing and the authors are encourage to explain this better in method.

Validity of the findings

The work has been carefully done. And the interpretation is well balanced and supported by the data and explains the limitations of this study well, and conclusions are valid.

Comments for the author

In specific:
It would be more clear if the authors show two views of the complex structure and label all the components in Figure 1.
In Figure 3, it would be better to show a monomer to avoid confusion.

·

Basic reporting

The points are clearly described in professional English language.
Introduction , background and results are well presented.
Structure conforms to PeerJ standard.
Figures are relevent.
Raw data supplied.

Experimental design

Original study within scope of the journal.
research questions are well addressed, and meaningful.
Methods described with sufficient detail.

Validity of the findings

Data is robust, statistically sound.
Conclusion well stated.
Speculation needs further experimental data from other researches.

Comments for the author

Evolution relies on mutation. For those notorious viruses such as Ebola virus which caused many deaths last year, the formidable mutagenesis ability endows the virus to fight back and escape any effective treatment. Increasing structural evidences provided by experimental techniques enabled analysis of antibody-based treatment to these viruses, yet it remains difficult to characterize the vast conformational repertoire of future mutants that resistant to current antibodies. Therefore, computational modeling and simulation techniques become useful for a detailed molecular understanding of biomolecular structures as well as prediction and treatment. Here the authors presented a watch list for Ebola virus escape mutations, by using molecular modeling to estimate the stability free energy change of mutations involving in the virus envelope glycoprotein GP and the known antibody KZ52, as a beginning. Other than conventional methods, they applied molecular simulation prior to molecular modeling, which reflects the intrinsic dynamics of biomolecules and overall reduces the statistics bias. The resulting list covers 34 mutations on GP2 subunit, in which one mutation has been identified in humans, suggesting the feasibility of this method.
In summary, this intriguing study is well designed and informative. The method described here can be applied to other targets as well in the future, if more experimental data support that the predicted mutations indeed disrupt the interaction between GP2 and KZ52. It's suitable to be accepted by PeerJ without further revision.

---

## Round 0.2 · accepted · Accept

· Academic Editor

Accept

We are pleased to inform you that your revised manuscript has been accepted for publication.